# Consumer Marketing Strategy and E-Commerce in the Last Decade: A Literature Review

**Albérico Rosário** [1,*] and **Ricardo Raimundo** [2]

1    GOVCOPP-Governance, Competitiveness and Public Policies, IADE—Faculdade de Design, Tecnologia e Comunicação, Universidade Europeia, 1500-210 Lisboa, Portugal
2    ISEC Lisboa, Instituto Superior de Educação e Ciências, 1750-142 Lisboa, Portugal; ricardo.raimundo@iseclisboa.pt
*    Correspondence: alberico@ua.pt

**Abstract:** E-commerce is deemed as the sale and purchase of goods and services through the internet in exchange for money and data transfer to complete the transactions. E-commerce is at the forefront of transforming marketing strategies, based on new technologies, and facilitates product information and improved decision-making. In this way, marketing strategy increasingly require large amounts of information to better understand client needs, which raises the question of choosing the right marketing strategy to better fit consumer expectations. This literature review aims to shed light on both the recent growth of e-commerce literature and its interplay with consumer marketing strategy. Extant research has examined this change in human interaction due to social network building, mostly through the themes of online marketing and social media marketing, also comprehending issues such as cost efficiency, information quality and trust development towards online shopping. Nevertheless, existing research has not shown in full all the research streams, how they interact with each other and its potential knowledge development. Thus, a literature review on consumer marketing strategy for e-commerce in the last decade is opportune. This paper aims to identify research trends in the field through a Systematic Bibliometric Literature Review (LRSB) of research on marketing strategy for e-commerce. The review includes 66 articles published in the Scopus® database, presenting up-to-date knowledge on the topic. The LRSB results were synthesized across current research subthemes. The following findings are presented: Amidst the current competitive global business environment, companies tend to respond with strategies for e-commerce and online businesses that resort to e-commerce platforms and social networking to better understand consumer needs, facilitate consumer marketing strategy and share innovative information. The originality of the paper relies on its LRSB method, together with extant review of articles that have not been categorized so far.

**Keywords:** consumer; marketing strategy; social media

## 1. Introduction

The digitalization of information and non-information products, due to technological developments and internet growth, has caused companies to rethink their marketing strategies. Competition has increased due to the creation of an internet-enabled marketplace that competes with the physical marketplace [1]. Consequently, companies have integrated the electronic market in their strategies to increase visibility and access the global market, leading to the growth of electronic commerce (E-commerce) [2]. E-commerce refers to the sale and purchase of goods and services through the internet, with the transfer of money and data to complete the transactions [3]. E-commerce platforms facilitate product information discovery that enables comparisons and decision-making [4]. They aim to replicate consumer in-store experiences and interactions to influence purchasing decisions [5]. Therefore, interactive marketing is significant in the internet-enabled market environment. Consumer marketing strategies, in this case, involve improved engagement and provision

of information resources to build knowledge and understand individual needs. Given the rapid increase in and sharing of information in online environments, companies struggle with identifying the most effective engagement and marketing strategies that align with consumer expectations and knowledge levels [6]. In this way, this piece of literature aims to analyze the growth of e-commerce in the last decade, and its interplay with consumer marketing strategy by evaluating the rapid changes and developments, as well as potential solutions. We achieve this goal through the identification of research trends with a Systematic Bibliometric Literature Review (LRSB), which comprehends 66 articles published in the Scopus® database. The ensuing results are synthesized by subthemes and findings are presented.

## 2. Materials and Methods

This article intends to identify the key developments in e-commerce and the corresponding consequences on marketing strategies in its state-of-the-art literature. This goal is attempted through a Systematic Bibliometric Literature Review (LRSB), which involves 66 articles published in Scopus® database. The LRSB method provides originality to the paper, along with an existing review of articles that have not been reviewed up to date in order to synthesize representative data on e-commerce and consumer marketing strategy and develop insightful perspectives and frameworks. Torraco [7] defines LRSB as a form of research that involves identifying an appropriate topic or problem, searching and retrieving relevant information sources to analyze and integrate literature to improve understanding. An article review is a fundamental method of knowledge development for both mature and emerging topics. Unlike other methodology, integrated reviews combine findings from multiple studies to place the study topic in context by summarizing knowledge and providing recommendations, as well as highlighting knowledge gaps for future studies [8]. Given the rapid growth of e-commerce, a comprehensive review can provide the information needed to develop and implement appropriate strategies to increase a company's competitiveness, performance, and productivity.

An integrated literature review involves thorough screening and selection of information sources to ensure validity and accuracy of the data interpreted and presented [9–12]. Therefore, this research adhered to the five scientific steps of conducting a literature review proposed by Russell [13]: formulating the question, conducting a literature search, evaluating data, analyzing data, and interpreting and presenting the results.

The database of scientific articles used was SCOPUS, the most important peer-review in the academic world. However, we acknowledge that the study has the limitation of considering only the SCOPUS database, excluding the other academic bases [9–12]. The bibliographic search includes peer-reviewed scientific articles published up to March 2021.

The keywords used include 'consumer marketing strategy', 'marketing', 'internet marketing', and 'online commerce' to select titles and abstracts. The search was limited to the subject area 'business', publication dates in the 2011–2021 period, and the exact keywords 'e-commerce' and 'e-commerce' were used to narrow the search and identify sources with appropriate content that explicitly discusses the study topic.

Consequently, 66 final sources were identified for analysis and integration in the final research report (Table 1).

**Table 1.** Screening Methodology.

| Database Scopus | Screening | Publications |
| --- | --- | --- |
| Meta-search<br>First Inclusion Criterion | keyword: Consumer<br>keyword: Consumer, Marketing strategy | 480,229<br>5.494 |
| Second Inclusion Criterion | keyword: Consumer, Marketing strategy<br>Subject area Business, Management and Accounting | 3.131 |
| | keyword: Consumer, Marketing strategy<br>Subject area Business, Management and Accounting and 2010–2021 period | 2.155 |
| Screening | keyword: Consumer, Marketing strategy<br>Subject area Business, Management and Accounting<br>Accounting and 2010–2021 period<br>Exactkeyword: Electronic Commerce, E-commerce<br>Published until March 2021 | 66 |

Source: own elaboration.

The 66 scientific articles are subsequently analyzed in a narrative manner to deepen the content and the possible derivation of common themes that directly answer the article's research question [9–12]. Of the 66 scientific articles selected, 50 are articles, 15 are conference papers, and one is a book chapter.

## 3. Publication Distribution

Figure 1 summarizes the published peer-reviewed literature on the study topic for the 2010–2021 period. The publications were categorized as follows: Proceedings of The International Conference On Electronic Business ICEB (6); Decision Support Systems (5); Journal of Retailing And Consumer Services (5); Asia Pacific Journal of Marketing And Logistics; Contemporary Management Research (2); Electronic Commerce Research And Applications; International Journal of Recent Technology And Engineering (2); Proceedings of The International Conference On E Business And E Government ICEE 2010 (2); with the following categories containing only one publication: 2nd International Conference On E Business And Information System Security EBISS 2010; 2nd International Conference On Artificial Intelligence Management Science And Electronic Commerce AIMSEC 2011 Proceedings; International Conference On Computing Mathematics And Engineering Technologies Invent Innovate And Integrate For Socioeconomic Development ICOMET 2018 Proceedings; Academy Of Marketing Studies Journal; Asian Academy Of Management Journal; Business Horizons; Education Business And Society Contemporary Middle Eastern Issues; Emerald Emerging Markets Case Studies; Indian Journal of Marketing; Industrial Marketing Management; Information Resources Management Journal; Information Systems Research; International Journal of Business Information Systems; International Journal of Culture Tourism And Hospitality Research; International Journal of E Entrepreneurship And Innovation; International Journal of Information And Management Sciences; International Journal of Information System Modeling And Design; International Journal of Logistics Systems And Management; International Journal of Networking And Virtual Organisations; International Journal of Production Economics; International Journal of Production Research; International Journal of Retail And Distribution Management; International Journal of Retail Distribution Management; Journal of Business Research; Journal of Cleaner Production; Journal of Direct Data And Digital Marketing Practice; Journal of Enterprise Information Management; Journal of International Consumer Marketing; Journal of Internet Banking And Commerce; Journal of Textile And Apparel Technology And Management; Journal of The Academy Of Marketing Science; Journal of Theoretical And Applied Electronic Commerce Research; Knowledge Based Systems; Lecture Notes In Business Information Processing; Managing Information Resources And Technology Emerging Applications And Theories; Proceedings International Conference On Management Of E Commerce And E Government ICMECG 2014; Proceedings 3rd International Conference On Data Science And Business Analytics ICDSBA 2019; Proceedings of 2019

International Conference On Information Management And Technology ICIMtech 2019; Service Industries Journal; Tourism Economics.

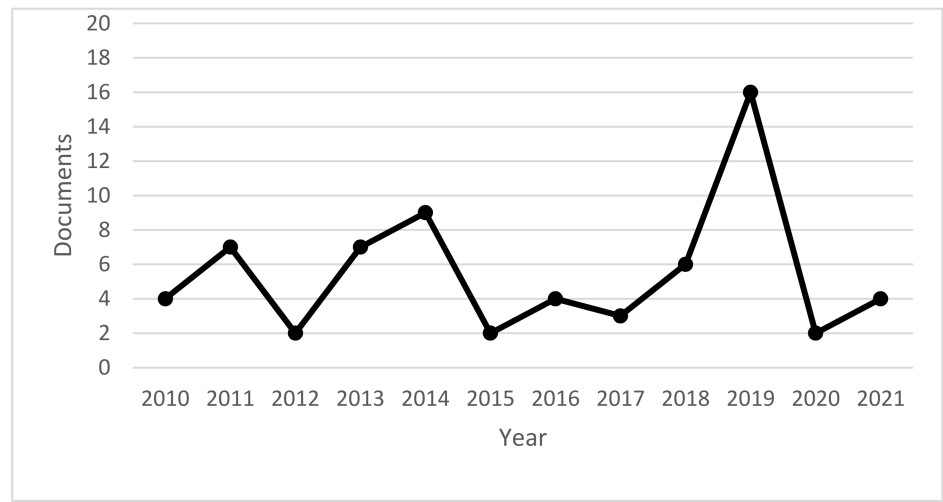

**Figure 1.** Documents by year. Source: own elaboration.

Interest in the subject has varied over time.

In Table 2 we analyze the Scimago Journal & Country Rank (SJR), the best quartile and the H index by publication.

**Table 2.** Scimago journal & country rank impact factor.

| Title | SJR | Best Quartile | H Index |
|---|---|---|---|
| Journal of the Academy of Marketing Science | 5.510 | Q1 | 170 |
| Information Systems Research | 3.510 | Q1 | 159 |
| Journal of Retailing and Consumer Services | 3.180 | Q1 | 136 |
| International Journal of Production Economics | 2.410 | Q1 | 185 |
| Business Horizons | 2.170 | Q1 | 87 |
| Journal of Business Research | 2.050 | Q1 | 195 |
| Industrial Marketing Management | 2.020 | Q1 | 136 |
| Journal of Cleaner Production | 1.940 | Q1 | 200 |
| International Journal of Production Research | 1.910 | Q1 | 142 |
| Knowledge Based Systems | 1.590 | Q1 | 121 |
| Decision Support Systems | 1.560 | Q1 | 151 |
| Electronic Commerce Research and Applications | 1.180 | Q1 | 74 |
| Service Industries Journal | 1.180 | Q1 | 66 |
| Tourism Economics | 0.810 | Q1 | 58 |
| International Journal of Retail and Distribution Management | 0.730 | Q1 | 78 |
| Asia Pacific Journal of Marketing and Logistics | 0.600 | Q2 | 46 |
| International Journal of Culture Tourism and Hospitality Research | 0.570 | Q2 | 31 |
| Journal of Theoretical and Applied Electronic Commerce Research | 0.560 | Q2 | 30 |
| Journal of International Consumer Marketing | 0.480 | Q2 | 45 |
| International Journal of Logistics Systems and Management | 0.370 | Q2 | 31 |
| Journal of Enterprise Information Management | 0.280 | Q3 | 21 |
| Journal of Textile and Apparel Technology and Management | 0.270 | Q3 | 24 |
| Information Resources Management Journal | 0.260 | Q3 | 41 |
| Indian Journal of Marketing | 0.240 | Q3 | 10 |
| Asian Academy of Management Journal | 0.230 | Q3 | 14 |
| International Journal of Business Information Systems | 0.210 | Q3 | 15 |
| Lecture Notes in Business Information Processing | 0.210 | Q3 | 49 |
| Emerald Emerging Markets Case Studies | 0.200 | Q3 | 5 |
| Contemporary Management Research | 0.190 | Q3 | 2 |

**Table 2.** *Cont.*

| Title | SJR | Best Quartile | H Index |
|---|---|---|---|
| International Journal of E Entrepreneurship and Innovation | 0.180 | Q3 | 2 |
| International Journal of Networking and Virtual Organisations | 0.170 | Q4 | 19 |
| International Journal of Information System Modeling and Design | 0.160 | Q4 | 16 |
| International Journal of Information and Management Sciences | 0.130 | Q4 | 22 |
| Proceedings of the International Conference on Electronic Business ICEB | 0.120 | -* | 7 |
| Journal of Internet Banking and Commerce | -* | -* | 23 |
| International Journal of Recent Technology and Engineering | -* | -* | 20 |
| Education Business and Society Contemporary Middle Eastern Issues | -* | -* | 19 |
| Academy of Marketing Studies Journal | -* | -* | 15 |
| Journal of Direct Data and Digital Marketing Practice | -* | -* | 13 |
| 2011 2nd International Conference on Artificial Intelligence Management Science and Electronic Commerce AIMSEC 2011 Proceedings | -* | -* | 11 |
| Proceedings of the International Conference on E Business and E Government ICEE 2010 | -* | -* | 10 |
| 2010 2nd International Conference on E Business and Information System Security Ebiss 2010 | -* | -* | 7 |
| Proceedings 2014 International Conference on Management of E Commerce and E Government ICMECG 2014 | -* | -* | 4 |
| 2018 International Conference on Computing Mathematics and Engineering Technologies Invent Innovate and Integrate for Socioeconomic Development Icomet 2018 Proceedings | -* | -* | -* |
| International Journal of Retail Distribution Management | -* | -* | -* |
| Managing Information Resources and Technology Emerging Applications and Theories | -* | -* | -* |
| Proceedings 2019 3rd International Conference on Data Science and Business Analytics ICDSBA 2019 | -* | -* | -* |
| Proceedings of 2019 International Conference on Information Management and Technology ICIMtech 2019 | -* | -* | -* |

Note: * data not available. Source: own elaboration.

The Journal of the Academy of Marketing Science is the most quoted publication with 5.510 (SJR), Q1 and H index 170.

There is a total of 15 journals in Q1, 5 in Q2, 10 in Q3 and 3 in Q4. Journals with the best quartile of Q1 represent 31.25% of the overall 48 journal titles; best quartile Q2 represents 10.45%, best quartile Q3 represents 20.83%, and best quartile Q4 represents 6.25%. Finally, for 15 of the publications representing 31.25%, the data are not available.

As evident from Table 2, the majority of articles on Consumer Marketing Strategy in E-commerce in the last decade ranked in the Q1 best quartile index.

The subject areas covered by the 66 scientific articles were: Business, Management and Accounting (66); Computer Science (29); Decision Sciences (18); Social Sciences (11); Economics, Econometrics and Finance (9); Engineering (7); Arts and Humanities (6); Psychology (5); Environmental Science (3); Mathematics (2); Energy (1); Medicine (1).

The most quoted article was "Consumer participation in using online recommendation agents: Effects on satisfaction, trust, and purchase intentions", with 119 quotes published in the Service Industries Journal 1180 (SJR), the best quartile (Q) and with an H index of 66.

The published article focuses on the study of greater consumer participation in the use of a recommended product, which leads to more satisfaction, confidence and greater purchase intention.

In Figure 2 we can analyze the evolution of citations of articles published between ≤2010 and 2021. The number of quotes shows a positive net growth with an R2 of 81% for the period ≤2010–2021, with 2020 reaching 245 citations.

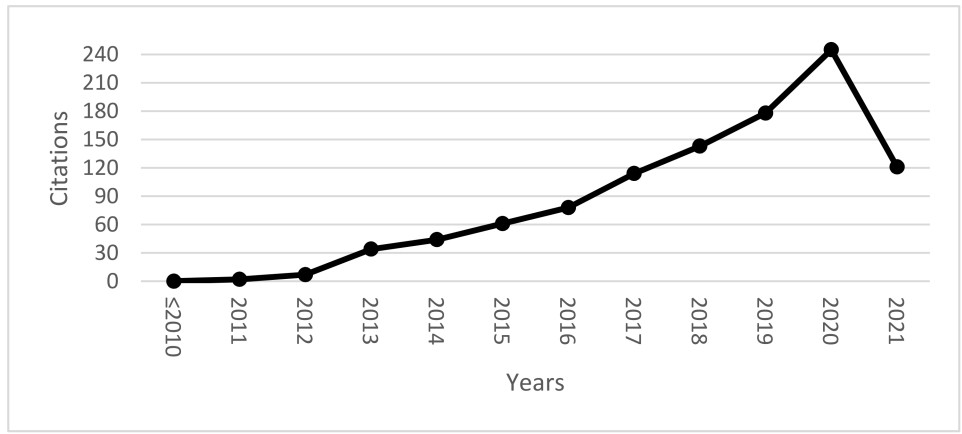

**Figure 2.** Evolution of citations between 2010 and 2020. (Source: own elaboration).

The h-index was used to ascertain the productivity and impact of the published work, based on the largest number of articles included that had at least the same number of citations. Of the documents considered for the h-index, 18 have been cited at least 18 times.

In Appendix A, the citations of all scientific articles from the ≤2011 to 2021 period are analyzed, with a total of 1027 citations. Of the 66 publications, 15 were not cited.

Appendix B examines the self-citation of the document during the period ≤2011 to 2021, 19 documents were self-cited 39 times, the article Measuring the effects of online-to-offline marketing by Chiang & Huang (2018) published in the Contemporary Management Research was cited 6 times.

In Figure 3, a bibliometric study was performed to examine the development of scientific information by the main keywords. The study of bibliometric outputs by the scientific software VOSviewe, aims to identify the main research keywords "Consumer", "Marketing strategy", and "E-commerce".

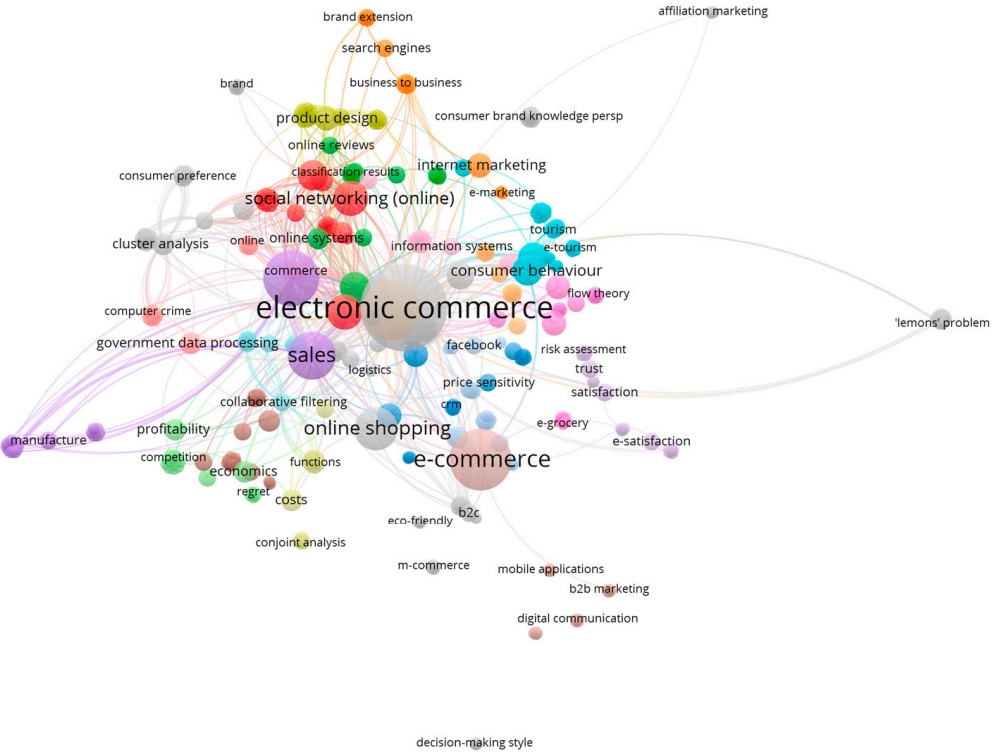

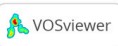

**Figure 3.** Network of all keywords. Source: own elaboration.

The research relied upon the studied articles on consumer marketing strategy on e-commerce in the last decade. The correlated keywords can be viewed in Figure 4, which allows the network of keywords that appear together/linked in each scientific article to be visualized and clarified, as well as understanding the topics studied to identify future research trends. Moreover, Figure 5 illustrates co-citations with a unit of analysis of cited references.

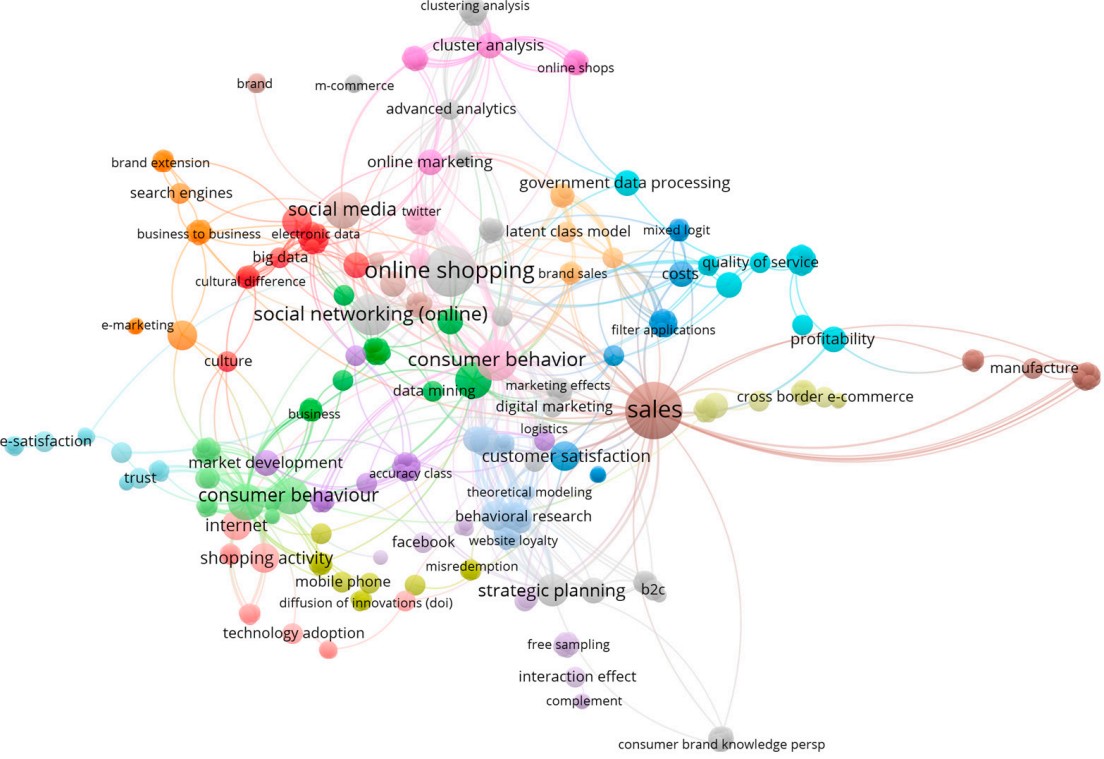

**Figure 4.** Network of linked keywords. Source: own elaboration.

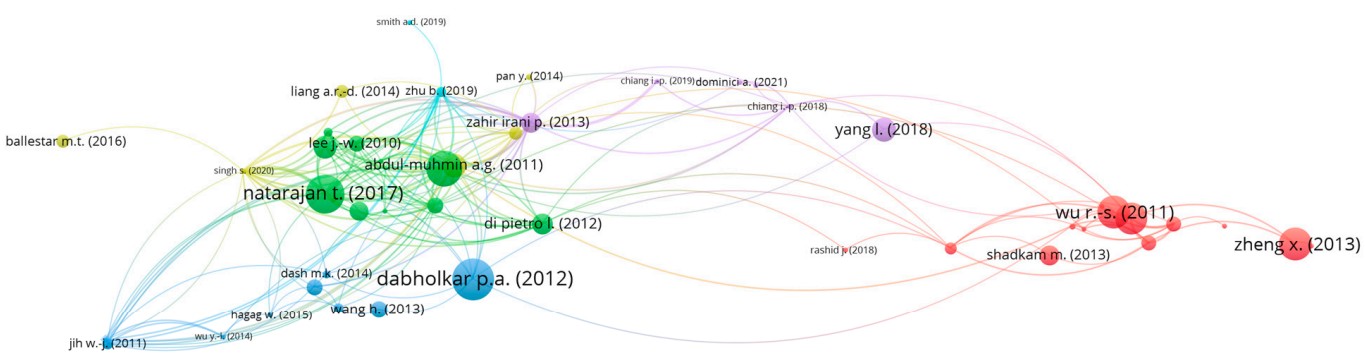

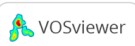

**Figure 5.** Networks of co-citation. Source: own elaboration.

## 4. Discussion

E-commerce involves the use of digital technologies in business to facilitate online sales and transactions. Kabugumila et al. [14] define e-commerce as the sale of goods and services using telecommunication and telecommunication-based technologies, such as the internet. Global technological trends and advancements have influenced consumers purchasing behaviors and intentions, encouraging online purchases and product information gathering due to perceived convenience and reduced costs [15]. These changes create the need for companies to establish online stores alongside physical stores to satisfy consumers' shopping needs and expectations [16]. In addition, the growth of e-commerce has further supported the development of retail platforms, such as Amazon and Alibaba. The variations in the nature of online business have led to categorizations that define e-commerce from various dimensions, including business-to-business (B2B), business-to-consumer (B2C), consumers-to-consumers (C2C), and government-to-business/consumer (G2B/C) [17]. The connectedness and ease of transactions promoted by modern technologies support these business activities, leading to local and international economic growth.

### 4.1. The Evolution of E-Commerce and Consumer Behavior

The E-commerce concept was initially coined to describe the processes of undertaking business activities electronically using technologies based on electronic funds transfer (EFT) and electronic data interchange (EDI). These technologies were developed in the late 1970s to enable information sharing and electronic transaction execution between organizations through invoices electronic purchase orders [14]. While these technologies laid the foundations for e-commerce, technological growth and the internet created online platforms that led to the development of the electronic business that combined data exchange and both monetary and non-monetary transactions. Ferrera and Kessedjian [18] indicate that the term e-commerce was developed in the 1990s following the development of new software and technologies that transformed the internet into a commercial environment. Wu and Hsieh [19] indicate that new technologies enabled organizations to share product and service information to influence consumer purchasing decisions. The interconnectedness promoted by the internet enables consumers to engage in various activities online, which creates a platform where companies offer awareness about themselves and their products [20]. However, e-commerce has undergone evolutions as technology advances [21]. Initially, companies used electronic technologies to facilitate transactions, such as sending documents after an order, while modern e-commerce involves purchasing and selling goods online [22]. E-commerce platforms are online marketplaces where consumers can view and purchase various products without traveling to physical stores, making them cheap and convenient.

The success of e-commerce is primarily supported by consumer involvement and engagement in internet-based environments. Hsu [20] indicates that it uses an experiential marketing strategy where consumers experience results from participating or observing an event that becomes stimuli and drives purchasing thoughts. A customer marketing strategy focuses on consumer senses, emotions, ideas, and behaviors depicted through online interactions. Yen [23] further explains that consumer behavior and purchasing intentions have shifted due to the growth of the internet and internet-based e-commerce platforms. For instance, nowadays, clients are concerned about organizational reputation, information satisfaction, and extended benefits. As a result, companies combine e-commerce and social media to maximize information distribution and increase brand awareness and visibility.

While social media provides information regarding a brand and the offerings, the e-commerce platforms create a platform where users can view all products and related information for decision-making. Yen [23] explains that well-known companies minimize uncertainties and build the initial trust needed to make online purchasing decisions. Unlike in traditional marketing, the evolution of e-commerce has made it possible for consumers to access necessary information about companies, their products, and business practices from websites and customer reviews and recommendations [24]. Therefore, e-commerce has

significantly promoted and is dependent on electronic word-of-mouth (eWOM) to attract and maintain existing and new customers [25]. Therefore, enhancing satisfaction through improved services, quality products, engagement, and experiences is fundamental to the success of businesses operating in online business environments. With data technologies, companies can track consumer purchasing activities and demographic features to match products and services to consumer needs and expectations.

### 4.2. The Role of Websites in E-Commerce and Consumer Marketing

In e-commerce, the processes of buying and selling products and services occur online. Therefore, e-commerce websites are online portals where these transactions occur, enabling transfer of goods ownership and monetary and information transfer. They are online shops where consumers select products and services and follow guidelines to pay and checkout awaiting delivery. Hagag et al. [26] indicate that companies should ensure that the web design and interface features, such as language, color, contrast, layout and positioning, size and shape, and texture, are appropriate for enhancing efficiency and user-friendliness. In addition, aesthetics, consistency, clarity, concision, and responsiveness can increase visitors' willingness to interact with the various products featured and spend more time on the platform [27,28]. When designing websites, companies should evaluate and understand the significance of cultural values and perceptions of web design. In some cultures, symbols, language, colors, and navigation styles are based on cultural beliefs that influence interpretations of various characteristics. Hagag et al. [26] recommend conducting research using cultural frameworks such as Hofstede's five dimensions of culture variability and Hall's context model (1984) to understand the connection be-tween culture and web interface and their implications on consumer behaviors and preferences. For instance, Hall's context model (1984) categorizes cultures into low-context and high-context (p. 69). Consumers prefer unambiguous, explicit, or direct communication in low-context cultures, while high-context cultures prefer nonverbal communication [29]. These differences can influence the nature of language incorporated in an e-commerce website to ensure that it addresses the cultural preferences of the target consumers. Therefore, it is essential to conduct market research and clearly define target customers and their cultural values before designing a website to ensure efficiency and acceptance.

The primary benefits of e-commerce websites occur from their interactive and connectivity features. Jih [30] identifies websites as the primary interaction platform that enables companies to maintain interactions with existing and potential customers to influence purchasing decisions. In marketing initiatives, websites are tools used by companies and customers to obtain maximum value in the e-commerce market [31]. Both parties leverage the power of knowledge and information created and shared throughout the knowledge economy. Effective website design and management enables companies to acquire strategic value through customer relationships while consumers use the information for empowerment to establish their positions as value co-creators [32]. Therefore, unlike in traditional marketing, the use of websites in e-commerce has created an empowered consumer with the capability to influence marketing strategies and business practices.

Consequently, companies have to evaluate their relationships and success from the customer's perspective and implement consumer-centered strategies [33]. Technology advances such as social media, search engines, voice-over-IP technologies, and distributed databases have enabled innovations that enhance interactivity and personalization of consumer experiences [34]. In addition, the massive amount of information available online for consumers' knowledge development and awareness has increased alternatives and ensured access to quality products and services. This new development has increased competition and prompted the need for companies to pay attention to consumer needs and demands, making them powerful components of business strategies and processes. Therefore, the marketing and business strategies implemented in the e-commerce environment are mutually beneficial to consumers and companies.

Additionally, e-commerce eliminates geographical barriers to facilitate global connections and reduce transaction costs. With e-commerce websites, companies can communicate information about their products and services with local and international consumers, making them essential communication and connection tools. For instance, Daries et al. [35] explain that e-commerce websites are used as primary information sources in the tourism industry, where tourists navigate the sites for information on services offered, accommodation, transport, and other tourism resources within the target area. Most consumers nowadays demand active experiences, which companies achieve through customer support and engagement [36]. They share ideas online, which are incorporated when developing innovative new products and services to increase their sense of ownership and connection with the brand, consequently leading to brand loyalty and frequent purchases. Jih [30] recommends that organizational decisions and operations be based on customer knowledge and a systematic understanding of customer relationships. Conducting consumer research using modern ICT tools facilitates customer profiling and understanding of multiple behavioral dimensions that influence purchasing intentions and behaviors [37]. Therefore, the growth of e-commerce contributes to the worldwide interconnection of firms and potential consumers, leading to an interdependent relationship where each party's actions and decisions influence the other.

E-commerce websites play a significant role in connecting small and medium-sized enterprises (SMEs) to global markets, increasing revenues and growth rate. Due to limited resources, SMEs are often limited to local markets while multinationals explore overseas opportunities. Fan [38] explains that cross-border e-commerce (CBEC) enables online imports where consumers in a particular country purchase foreign goods at lower prices. With these technologies, SMEs can create websites or rent online storage space in e-commerce websites such as Amazon and Tmall to reach potential global customers. Ballestar et al. [39] define e-commerce as a "new system with no rules" where large and small network communities create affiliates to obtain value. Global internet connectivity has no gatekeepers and operates at low costs, enabling small companies to market their products and services [40]. However, SMEs need collaborations with other organizations to enhance marketing, delivery, packaging, and payment services and optimize the opportunities created in the worldwide markets [41]. These developments have reduced the trade barriers existing in traditional e-commerce that often promote monopoly and growth of large companies, while SMEs control a limited market base. Fan [38] estimated that the global B2C e-commerce would grow between 2014 and 2020 from US$230 billion to US$994. While many of these revenues are earned by large corporations, CBEC enables SMEs to compete in the global markets by maximizing the opportunities created by the cross-border e-commerce ecosystem. Therefore, SMEs should understand the export growth model to identify and implement strategies that enhance their performance in local and international markets. By considering navigation, information, and visual designs, SMEs can establish e-commerce websites that compete with other companies operating within the internet-enabled business environment.

### 4.3. Consumer Behavior and Purchasing Decision in E-Commerce

Online shoppers' behaviors and intent to buy are influenced by the availability of product-related information and costs. The internet allows consumers to compare product prices, access consumer reviews and recommendations, and search for products offering various products they need [42]. E-commerce utilizes software technology to create consumer profiles by understanding their interest and preferences to provide appropriate product recommendations. These technologies enable first to overcome problems arising from the availability of excess information online, which can be frustrating and confusing to consumers searching for brand data for decision-making [43]. The e-commerce platforms include design features that allow two-way communication with consumers to provide support and in-formation resources regarding products of interest [41]. In addition, consumer data collection facilitates personalization by tracking browsing and consumption

patterns and initiating consumer inputs for improved experiences and service quality. Some marketers use recommendation agents (RAs) to increase consumer interactions and guide consumers through information search and purchasing decision-making. In consumer marketing, customer input is essential in creating and implementing promotional campaigns [44]. Modern internet-based business environments require optimal consumer engagement in all business practices to ensure that strategies implemented match consumers' standards and appeal to target audiences. Therefore, e-commerce companies such as eBay, Amazon, and Yahoo have integrated recommendation technologies in their websites to elicit consumer preferences through two-way communication, where they share opinions and request information to aid decision-making. These developments highlight the significance of product information in influencing purchasing intents and online behaviors.

In addition to RA technologies, e-commerce websites are adopting augmented reality (AR) to improve consumer experience and influence purchasing intentions. Qin et al. [45] indicate that AR technologies enable online shoppers to visualize and evaluate products and services before buying. For example, consumers can use AR to virtually furnish actual rooms when purchasing furniture from an e-commerce platform by selecting the needed furniture from an online showroom and visually placing it in the targeted real space [46]. The Amazon AR View is an example of AR technologies that alleviate consumer concerns on the appropriateness of online purchased products and their suitability in the real space. Therefore, while RA provides information resources, AR creates a technology-enabled visual experience that enables consumers to view and experience the targeted product. These technologies continue to evolve e-commerce and influence online purchasing intentions.

Consumer marketing strategy in e-commerce uses social networking sites to create brand awareness and influence consumer behaviors and buying decisions. Social media enables companies to generate online sales by building a fanbase through online followers who share their products, profile, and brand online. Di Pietro and Pantano [47] indicate that low connection costs allow social networks to connect multiple actors within the market who share standard product and service knowledge and link firms to their target consumers. The global connection facilitates consumer marketing since marketers can gather consumer demographics data to build customized promotional messages and direct advertising [48]. It creates a virtual space where online users and communities publish, share, and retrieve information regarding products and services [49]. In popular platforms such as Facebook, consumers and marketers can share information in text, photos, videos or initiate discussions on specific issues or brands of interest. Data gathered from such platforms can be exploited to create targeted marketing or generate innovative ideas on products or services based on consumer needs and standards. In addition, social networks allow consumers to share experiences and judgments regarding a brand, which other potential clients use as sources of reliable information [50]. Therefore, consumer reviews and recommendations are a form of electronic word of mouth that can direct online shoppers to a company's e-commerce platform [51]. Although the growth of social media reduces the company's control over information shared online, it creates opportunities for building brand awareness through brand-consumer and consumer-consumer relationships and inter-actions. It promotes relationship marketing, which further leads to loyalty and trust, and consistent sales and revenues.

The familiarity and popularity of online shops influence purchasing decisions and the growth of e-commerce platforms. According to Okamoto's [52] research, online shoppers prioritize popular online stores and choose well-known ones. Various factors influence a store's popularity, including website quality, shopping experiences, and product and service quality. Given the uncertainties and risks involved in online businesses, consumers pay attention to a website's security, usability, and privacy features [50]. Other attributes include delivery services, convenience, prices, trust, and product varieties. When these demands are met, consumer experiences and satisfaction will be higher, encouraging frequent purchases that build long-term relationships. Okamoto [52] indicates that consumers

expect to acquire value and benefits from online shopping unavailable in traditional stores, such as better prices, varieties, and convenient shopping processes. Thus, e-commerce growth is significantly de-pendent on consumers' perceived value attained from completing certain online transactions [53]. However, consumers prioritize different attributes when determining shopping experiences. The diversity requires companies and marketers to understand consumers' online behaviors and patterns to customize services and improve delivery [54]. Therefore, incorporating modern data gathering and analysis technologies in e-commerce websites is critical. Quality and affordable services and products and trustable purchasing and delivery processes can help build brand reputation that leads to consumer advocacy and brand popularity.

Consequently, the online-to-offline (O2O) e-commerce model has become a popular operational strategy that maximizes consumers' benefits and value. The increased product and brand information has prompted consumers to shift to multichannel e-commerce to source products online and offline [55]. For example, some customers use e-commerce websites to source product data, such as prices and reviews, and use it in offline purchasing. Wang et al. [56] explain that multichannel shoppers evaluate factors, such as prices and brand reputation online but visit physical stores to evaluate and experience product quality before buying. Others avoid online shopping due to perceived risks of scamming and data theft, opting for traditional commerce that enables in-person transactions [57]. Therefore, adopting the O2O business model can help tap online and offline buyers by ensuring access to diverse information and experiences through online and offline stores, increasing satisfaction [58]. Although online platforms provide various goods, attractive prices from multiple sellers, good product information, and convenient transactions, they do not offer real product experiences and complete consumer services [54]. Neither online shopping platforms nor traditional e-commerce can provide consumers with maximum, creating the need for cross-channel buying. Some customers prefer in-person buying in physical stores, although they conduct online brand searches for purchasing information. Therefore, despite the shift to online commerce, firms should maintain physical stores to ensure they cater to diverse consumer needs and improve experiences to build trust and loyalty.

Risk perception constitutes a significant challenge in e-commerce that reduces consumer willingness to buy online. Gao and Hu [59] explain that about 90% of perceived risks occur from six dimensions that include "performance risk, physical risk, economic risk, psychological risk, social risk, and time/convenience risk" (p. 2188). Since online buying does not involve product trials, buyers fear that the products may not meet the desired quality and functionality and avoid buying. Other issues affecting online transactions include privacy and quality concerns and incidental risks [60]. Online transactions require clients to fill in their financial information, which causes some to fear the risk of financial loss [61]. For example, embezzling credit cards via the internet can lead to money loss. In addition, some goods might be destroyed during delivery. These issues cause perceived risks that discourage internet users from engaging in e-commerce [57].

To summarize, companies need to prove appropriate risk reduction strategies to attract online shoppers and completion on online transactions. These issues reflect the significance of Okamoto's [52] argument that online shoppers are more likely to prioritize well-known and reputable e-commerce stores. The popularity and familiarity reduce perceived risks since consumers understand and respect the company's business practices, product quality, and delivery processes. While offering quality products and services is essential, marketers should also integrate consumer-centered initiatives to build trust and relationships that increase consumers' attitudes and perceptions.

### 4.4. The Use of Big Data in Consumer Marketing and E-Commerce

The rapid development of science and technology has significantly influenced information flow in e-commerce and created the need for analytics tools for better optimization. The internet has created a free platform where data is shared and transferred from individual internet users and brands, leading to data volumes skyrocketing. E-commerce companies

use big data to improve decision making, product quality, performance, and operations that create a competitive advantage. Shi and Xu [62] explain that the technologies have gained significance due to the consumption shift from a product orientation to a consumer orientation. Unlike in traditional commerce, contemporary business environments require organizations to implement strategies and product designs that are customer-centered, following technological changes that have created an empowered global consumer base [63]. A competitive product design utilizes science and technology to create elements that transform consumer feelings, values, culture, and consciousness to enhance experiences and satisfaction [64]. With big data technologies, firms gather and analyze customer data to customize and personalize products and services to match product quality with customer demands. In addition, it enables marketers to study patterns and project future changes to create and implement timely and convenient marketing strategies that address various rapidly occurring concerns that may undermine e-commerce business models [65]. For instance, the internet allows anyone to publish a post on a brand, company, or product. While this information flow can generate sales through reviews and recommendations, it can also lead to misinformation and false accusations that might damage an organizational reputation and public image. Therefore, big data technologies can help track and analyze such information to understand its origin, basis, and potential consequences, to enable marketers and other communication and public relation professionals to curate appropriate messages to mitigate the problem.

Due to the increased number of e-commerce platforms and information influx, companies are adopting recommender systems (RS) to provide personalized recommendations. Beladev et al. [66] define RS as information filtering applications used to create customized recommendations for products and services to online shoppers. The primary goal of using recommender systems in e-commerce is to converse online browsers into buyers by increasing their exposure to quality products and services based on their search and offering suggestions for additional related products [65]. These technologies enhance consumer experiences by improving navigation and easy access to product and service varieties. RS increases revenues and enhances a company's marketing techniques since consumers are guided towards the specific products or services needed. Beladev et al. [66] indicate that RS is especially successful in product bundling marketing strategy, which offers two or more items packaged together at a relatively lower price than buying each individually. Using RS, in this case, benefits the company and consumers in that it widens their purchasing scope while increasing incomes and profits [33]. The RS can direct consumers towards product bundles consisting of products that would have otherwise not been bought. Product prices significantly influence purchasing decisions and repeat purchases. Abdul-Muhmin [36] indicates sustained growth for the e-commerce industry is dependent on loyal patrons who are defined by repeat purchase intentions. The internet grants consumers access to product information, including quality, sustainability, and prices [67]. They use this information to compare competing products and select those with competitive prices. Similarly, companies and marketers can optimize online data to create pricing strategies that attract and maintain target customers to enhance competitiveness in the industry and a positive brand image.

The contemporary dynamic and competitive business environment require a better understanding and forecasting of consumer demands. Unlike in conventional commerce, where companies achieved competitiveness by lowering production costs and improving product quality, e-marketplace's focus has shifted towards standardization of products and services due to the increased need for customizations [68]. Information Technologies and data science empower manufacturers with tools that enhance the understanding of consumer demands. For example, companies use Radio Frequency Identification Tags (RFID) to acquire real-time inventory data and spatial mobility, which are used to project product demand [64]. They use data analytics to understand complex relationships between various business dimensions to improve processes, performance, and product quality. Informational resources have become fundamental elements in business strategy since

they enable companies to understand their interdependent relationships with customers in the current customer-oriented business models [69]. Through big data technologies, firms have access to data from multichannel that can be used to improve the supply chain and exploit untapped business information. Chong et al. [64] indicate that 53% of posters on Twitter recommend products or brands, and 48% of Twitter users interacting with the tweets follow the recommendation. These statistics highlight the power of user-generated content in influencing consumption decisions. Therefore, using Twitter analytics tools can enable firms to forecast product demand and implement appropriate strategies. Thus, big data technologies are critical tools for e-commerce industry growth and sustainability.

Global economic growth and increased disposable incomes have led to emerging markets, creating opportunities for companies to expand worldwide operations. However, understanding processes in these new foreign markets requires a comprehensive analysis of economic, social, security, and cultural data to evaluate the general business environment and determine consumption behaviors [70]. The authors projected economic growth of the e-commerce industry in India between 2012 and 2016 from USD1.6 billion to USD8.8 billion. This growth identifies India as an emerging economy projected to rank as the fifth largest consumer market globally by 2025. Despite these promising statistics, companies need data to understand India's business environment and operational issues required to exploit the emerging opportunities. Big data can provide insights into consumer demographics, value creation, and resource management to increase performance and create practical entrepreneurial strategies. Liu et al. [70] acknowledge the challenges experienced by manufacturers penetrating new markets and indicate that strategic decisions directly affect sales performance and marketing strategy. Dominici et al. [68] recommend using local agents, such as distributors, to facilitate effective penetration to these emerging markets. However, a company's supply chain should integrate data science and information technologies to ensure appropriate data-driven decision-making [70]. The information flow and economic changes facilitated by economic growth lead to changing online consumer behaviors, highlighting the need for adequate tracking tools. Therefore, big data should be used alongside active marketing campaigns to increase brand visibility and implement data-driven strategies that forecast current and future product demand and market performance.

Big data facilitates customer marketing strategy by enabling companies and marketers to identify and understand their segments and their content preferences to enhance positive responses. Segmentation allows companies to divide target markets into small defined groups consisting of shared characteristics, such as needs, demographics, geographical location, and interests. Arbi Siti et al. [71] found that effective marketing content should be more visual and include a personal touch to make it appealing and relatable to target markets and arouse purchasing desires. Big data avail large amounts of consumer data that ensure organizational strategies and marketing information matches the rapidly changing consumer needs and interests [72]. Companies need to understand that technological changes, information richness, economic changes, and education, among other factors, influence consumer needs and interests [73]. Therefore, an individual's expectations today might change tomorrow due to varying developments and lifestyle changes. Big data helps companies track these changes by analyzing activity-based data, social networks, and sentiment data such as brand associations, reviews, and comments. Arbi Siti et al. [71] identify three consumer segmentation dimensions: demographics, behavioral, and marketing content dimensions.

### 4.5. Modes of Segmentation

In the vein of the aforementioned rationale of this piece of literature, it is key to understand the diverse forms of segmentation that come up with the corresponding ways of responding to this new, digital and global environment.

### 4.5.1. Demographic Segmentation

Market segmentation based on demographic factors enables companies and marketers to understand target groups' needs, wants, and consumption levels. Some demographic factors considered include "age, gender, sex, marital occupation status, family size, family life cycle, income, education, religion, race, and nationality" (p. 202) [71]. Gender differences influence individual behavior, psychology, and needs. Therefore, marketers need to create and distribute promotional messages that address the concerns of each gender group based on the company's target market segment [72]. In addition, age significantly influences online consumer behaviors. For instance, online purchases are more prevalent among millennials and Gen Z due to their tech knowledge and familiarity, which leads to more straightforward navigation of e-commerce websites and transaction processes. Zhu et al. [74] indicate that Generation Y (Millenials) make up the largest consumer group in the e-commerce marketplace, and possess a more substantial purchasing power. Dominici et al. [68] indicate that young generations are energetic and spend most of their time creating and sharing user-generated content on social media to develop and influence new shopping styles. However, their loyalty to a brand is uncertain due to increasingly changing preferences and choices, challenging organizational attempts to achieve consumer retention. Big data provides practical solutions to this challenge by delivering analytical re-sources that understand people's consumption behaviors and project potential future changes. The technologies promote innovative strategies that accommodate turbulent market changes to maintain existing customers and attract new ones.

### 4.5.2. Behavioral Segmentation

Behavior characteristics determine an individual's actions towards obtaining and using goods and services. Arbi Siti et al. [71] divide behavior segments into two; access intensity and time. Access intensity involves the frequency of an individual's internet access. Internet users can be classified as light, medium, or heavy users depending on the amount of time spent online within a specified duration [75]. Light internet users spend minimal time online and use fewer social networking sites than medium and heavy internet users. Understanding the target consumers' access intensity enables companies to create and implement marketing strategies aligned with these groups [76]. For instance, heavy internet users spend a lot of time online, meaning they have more engagements and potential, and tech skills to withstand the online shopping process. Qian et al. [77] explain that people in the middle level of income and consumer education are likely to spend less time online due to the considerations of the time cost of online shopping. They have less free time due to economic and social responsibilities, such as jobs and family caretaking. Consumers who spend a lot of time online are likely to engage in e-commerce since they have access to multiple consumers and company-generated content regarding brands, specific products, or trendy styles that appeal to them. Wang et al. [78] indicate that online shoppers tend to spend a lot of time inspecting and selecting promising products before placing them on the e-commerce shopping cart and checking out. Big data technologies are incorporated in the e-marketplaces to trace consumers tracing behaviors to create groups based on access intensity and time spent online.

### 4.5.3. Marketing Content Segmentation

Companies use marketing to educate target consumers and create awareness regarding the company and its associated products and services. Therefore, marketing content is fundamental in communicating an organization's value and influencing consumers purchasing decisions [71]. Various factors considered in marketing content include originality, relevance, timeliness, simplicity, and call to action. The promotional messages distributed to online shoppers should be original to ensure differentiation from competitors, new, and suitable to enhance trust and build online relationships. Pan [79] found that many internet users used e-commerce platforms to research products and not buy them due to quality information and prices. While companies tend to commercialize the

convenience granted by online environments and their appeal to consumers, they fail to provide products and services that guarantee additional valuable benefits [75]. For instance, some online products are more expensive than offerings in traditional commerce channels. These negative issues affect online users' perceptions of e-commerce and undermine trust. Consequently, e-marketing should fill this gap by providing accurate and reliable information on the benefits of using a company's e-commerce platforms and their variation from other platforms [76]. In this case, big data technologies can be used to gather and analyze consumer and market information to compare demands and expectations with the company's performance. As a result, marketers can identify knowledge and performance gaps and implement appropriate strategies to create attractive, educative, and entertaining information that builds consumer engagement.

## 5. Conclusions

Technological development has led to the digitalization of information and non-information products, encouraging firms to recreate their marketing and sales strategies. E-commerce is a significant development resulting from these technological changes that have significantly shifted commerce from traditional physical stores to internet-enabled marketplaces. Online businesses facilitate a consumer marketing strategy that is interaction and information-based to enhance efficiency, experiences, and satisfaction. For instance, e-commerce websites consist of design features that ensure responsiveness, clarity, consistency, and concision to enhance interactivity and engagement. They include colors, languages, navigation styles, and layouts that promote user-friendliness and match target consumers' expectations. E-commerce websites and social networking sites are jointly used to increase connectivity and interactivity. While the websites provide various products and services, SNSs provide two-way communication channels that enable consumer-to-consumer and consumer-to-company interactions. The communication facilitates sharing informational resources and developing innovative ideas and product designs that accommodate consumer demands and needs. In addition, e-commerce platforms integrate IT and big data technologies to promote the personalization and customization of user experiences. Software technologies, such as recommendation agents (RA) and recommender systems (RS), increase consumer interactions and provide product or service suggestions. Data analytical tools enable companies and marketers to track and analyze consumer behaviors and patterns and their implications on purchasing decisions. In the current competitive global business environment, understanding consumer perspectives and needs is critical in ensuring the success of e-commerce businesses.

**Author Contributions:** Conceptualization, A.R. and R.R.; methodology, A.R. and R.R.; software, A.R. and R.R.; validation, A.R. and R.R.; formal analysis, A.R. and R.R.; investigation, A.R. and R.R.; resources, A.R. and R.R.; data curation, A.R. and R.R.; writing—original draft preparation, A.R. and R.R.; writing—review and editing, A.R. and R.R.; visualization, A.R. and R.R.; supervision, A.R. and R.R.; project administration, A.R. and R.R.; funding acquisition, A.R. and R.R. All authors have read and agreed to the published version of the manuscript.

**Funding:** This research received no external funding.

**Institutional Review Board Statement:** Not applicable.

**Informed Consent Statement:** Not applicable.

**Data Availability Statement:** Not applicable.

**Acknowledgments:** We would like to express our gratitude to the Editor and the Referees. They offered extremely valuable suggestions or improvements. The authors were supported by the GOYCOPP Research Unit of Universidade de Aveiro and ISEC Lisboa, Higher Institute of Education and Sciences.

**Conflicts of Interest:** The funders had no role in the design of the study; in the collection, analyses, or interpretation of data; in the writing of the manuscript, or in the decision to publish the results.

## Appendix A

**Table A1.** Overview of document citations period ≤2011 to 2020.

| Documents | | ≤2011 | 2012 | 2013 | 2014 | 2015 | 2016 | 2017 | 2018 | 2019 | 2020 | 2021 | Total |
|---|---|---|---|---|---|---|---|---|---|---|---|---|---|
| Determinants of online food purchasing: The impact of socio- … | 2021 | - | - | - | - | - | - | - | - | - | - | 2 | 2 |
| A virtual market in your pocket: How does mobile augmented r … | 2021 | - | - | - | - | - | - | - | - | - | - | 1 | 1 |
| A multi-face! item response theory approach to improve custo … | 2019 | - | - | - | - | - | - | - | - | - | 2 | 1 | 3 |
| Generation Y consumer online repurchase intention in Bangkok … | 2019 | - | - | - | - | - | - | - | - | - | 4 | 3 | 7 |
| Garuda Indonesia new digital experience concept: Airline's eh … | 2019 | - | - | - | - | - | - | - | - | 1 | - | - | 1 |
| An exploratory study of cross border E-commerce (CBEC) in Ch … | 2019 | - | - | - | - | - | - | - | - | - | 3 | - | 3 |
| Understanding user-generated content and customer engagement … | 2019 | - | - | - | - | - | - | - | - | 1 | 9 | 6 | 16 |
| Cooperative behavior and information sharing in the e-commer … | 2019 | - | - | - | - | - | - | - | - | 1 | 13 | 1 | 15 |
| Measuring the effects of online-to-offline marketing | 2018 | - | - | - | - | - | - | - | - | - | - | 1 | 1 |
| The manufacturer's joint decisions of channel selections and … | 2018 | - | - | - | - | - | - | - | 6 | 5 | 14 | 16 | 41 |
| Advertiser's perception of Internet marketing for small and … | 2018 | - | - | - | - | - | - | - | - | - | 1 | - | 1 |
| Impact of flow on mobile shopping intention | 2018 | - | - | - | - | - | - | - | 2 | 8 | 18 | 9 | 37 |
| Predicting consumer product demands via Big Data: the roles … | 2017 | - | - | - | - | - | 1 | 5 | 11 | 18 | 23 | 12 | 70 |
| Understanding the intention to use mobile shopping applicati … | 2017 | - | - | - | - | - | - | 3 | 12 | 33 | 37 | 20 | 105 |
| The interplay between free sampling and word of mouth in the … | 2017 | - | - | - | - | - | - | - | 3 | 3 | 8 | | 15 |
| Recommender systems for product bundling | 2016 | - | - | - | - | - | | 3 | 8 | 5 | 5 | 4 | 25 |
| Consumer behavior on cashback websites: Network strategies | 2016 | - | - | - | - | - | 2 | 1 | 2 | 2 | 3 | 2 | 12 |
| Online store discount strategy in the presence of consumer 1 … | 2016 | - | - | - | - | - | 3 | 4 | 5 | 3 | 3 | - | 18 |
| Willingness to use fashion mobile applications to purchase f … | 2015 | - | - | - | - | - | | 1 | 1 | 1 | 3 | - | 6 |
| A framework for understanding the website preferences of Egy … | 2015 | - | - | - | - | - | 2 | - | - | - | - | - | 2 |
| The interaction effect on customer purchase intention in e-e … | 2014 | - | - | - | - | - | 2 | 3 | 1 | 4 | 2 | - | 12 |
| Bringing product and consumer ecosystems to the strategic fo … | 2014 | - | - | - | - | 1 | 1 | 3 | 2 | 2 | 1 | 1 | 11 |
| Information diffusion O2O model based on social learning | 2014 | - | - | - | - | - | | 1 | - | - | - | - | 1 |
| E-marketing under the adverse selection environment: Modela … | 2014 | - | - | - | - | - | 1 | - | 1 | - | 1 | - | 3 |
| Electronic consumer style inventory: Factor exploration and … | 2014 | - | - | - | - | - | 2 | 1 | 1 | 1 | - | - | 5 |
| Consumer Priorities in Online Shopping | 2014 | - | - | - | - | - | - | 1 | - | - | - | - | 1 |
| The role of sunk costs in online consumer decision-making | 2014 | - | - | - | 1 | 1 | 1 | 3 | | 2 | 3 | 1 | 12 |

**Table A1.** *Cont.*

| Documents | | ≤2011 | 2012 | 2013 | 2014 | 2015 | 2016 | 2017 | 2018 | 2019 | 2020 | 2021 | Total |
|---|---|---|---|---|---|---|---|---|---|---|---|---|---|
| Capturing the essence of word-of-mouth for social commerce: . . . | 2013 | - | - | - | 5 | 4 | 4 | 14 | 10 | 20 | 13 | 6 | 76 |
| From clicking to consideration: A business intelligence appr . . . | 2013 | - | - | - | 2 | 1 | 4 | 6 | 5 | 1 | | 1 | 20 |
| The antecedents of travellers' e-satisfaction and intention . . . | 2013 | - | - | - | 2 | 1 | 1 | 6 | 7 | 4 | 4 | 2 | 27 |
| Social commerce dimensions: The potential leverage for marke . . . | 2013 | - | - | 1 | 1 | 1 | 1 | 4 | 7 | 5 | 4 | 3 | 27 |
| An Analysis ofthe Existing Literature on B2C E-commerce | 2013 | - | - | | | | 2 | 4 | 1 | - | - | - | 7 |
| Online information product design: The infiuence of product . . . | 2013 | - | - | - | 2 | 3 | 4 | - | 2 | 7 | 1 | - | 19 |
| An empirical investigation of social network influence on co . . . | 2012 | - | - | 6 | 1 | 4 | - | 1 | 6 | 5 | 6 | 2 | 31 |
| Consumer participation in using online recommendation agents . . . | 2012 | - | 1 | 9 | 11 | 11 | 13 | 16 | 18 | 13 | 21 | 6 | 119 |
| Limitations of e-commerce in developing countries: Jordan ca . . . | 2011 | - | - | 1 | - | 5 | 4 | 6 | 4 | 3 | 2 | 1 | 26 |
| The analysis of B2C e-commerce implementation of the experie . . . | 2011 | - | - | - | 1 | - | - | - | - | 1 | - | - | 2 |
| Internet usage for travei and tourism: The case of Spain | 2011 | - | 2 | 1 | 1 | 2 | 2 | 2 | 1 | 2 | 5 | - | 18 |
| lnfluence ofsocial norrns, perceived playfulness and online . . . | 2011 | 1 | 2 | 4 | 5 | 10 | 10 | 11 | 11 | 12 | 13 | 7 | 86 |
| Customer segmentation of multi pie category data in e-commerc . . . | 2011 | 1 | - | 5 | 5 | 8 | 8 | 9 | 8 | 12 | 12 | 8 | 76 |
| lmpact of e-CRM on Website Loyalty of a Public Organizations . . . | 2011 | - | - | 2 | | 2 | 1 | 1 | | | 1 | 1 | 8 |
| Repeat purchase intentions in online shopping: The role of s . . . | 2011 | - | 2 | 1 | 6 | 3 | 8 | 3 | 5 | 2 | 6 | 3 | 39 |
| Typology of consumers' risk perceptions in online shopping: . . . | 2010 | - | - | - | - | - | - | - | - | 1 | 1 | - | 2 |
| The roles of demographics on the perceptions of electronic c . . . | 2010 | - | - | 4 | 1 | 4 | 1 | 2 | 3 | - | 3 | - | 18 |
| | Total | 2 | 7 | 34 | 44 | 61 | 78 | 114 | 143 | 178 | 245 | 121 | 1027 |

## Appendix B

**Table A2.** Overview of document self-citation period ≤2011 to 2021.

| Documents | | ≤2011 | 2012 | 2013 | 2014 | 2015 | 2016 | 2017 | 2018 | 2019 | 2020 | 2021 | Total |
|---|---|---|---|---|---|---|---|---|---|---|---|---|---|
| A multi-face! item response theory approach to improve custo . . . | 2019 | - | - | - | - | - | - | - | - | - | 1 | - | 1 |
| Garuda lndonesia new digital experience concept: Airline's eh . . . | 2019 | - | - | - | - | - | - | - | - | 1 | - | - | 1 |
| Understanding user-generated content and customer engagement . . . | 2019 | - | - | - | - | - | - | - | - | 2 | - | - | 2 |
| Cooperative behavior and information sharing in the e-commer . . . | 2019 | - | - | - | - | - | - | - | - | 3 | - | - | 3 |
| Measuring the effects of online-to-offline marketing | 2018 | - | - | - | - | - | - | 2 | 1 | 2 | 1 | - | 6 |
| Predicting consumer product demands via Big Data: the roles . . . | 2017 | - | - | - | - | - | 1 | | 1 | - | - | - | 2 |
| Understanding the intention to use mobile shopping applicati . . . | 2017 | - | - | - | - | - | - | 1 | - | 1 | - | - | 1 |
| The interplay between free sampling and word ofmouth in the . . . | 2017 | - | - | - | - | - | - | - | - | 1 | - | - | 1 |

**Table A2.** *Cont.*

| Documents | | ≤2011 | 2012 | 2013 | 2014 | 2015 | 2016 | 2017 | 2018 | 2019 | 2020 | 2021 | Total |
|---|---|---|---|---|---|---|---|---|---|---|---|---|---|
| Recommender systems for product bundling | 2016 | - | - | - | - | - | - | 1 | - | 1 | - | - | 2 |
| Consumer behavior on cashback websites: Network strategies | 2016 | - | - | - | - | 1 | - | 1 | 1 | - | - | - | 3 |
| Bringing product and consumer ecosystems to the strategic fo . . . | 2014 | - | - | - | 1 | - | - | - | - | - | - | - | 1 |
| Electronic consumer style inventory: Factor exploration and . . . | 2014 | - | - | - | - | - | 2 | 1 | - | 1 | - | - | 4 |
| Consumer Priorities in Online Shopping | 2014 | - | - | - | - | - | - | 1 | - | - | - | - | 1 |
| From clicking to consideration: A business intelligence appr . . . | 2013 | - | - | - | - | - | 2 | - | 1 | - | - | - | 3 |
| An Analysis ofthe Existing Literature on B2C E-commerce | 2013 | - | - | - | - | - | 1 | - | - | - | - | - | 1 |
| An empirical investigation of social network influence on co . . . | 2012 | - | - | 4 | 1 | 1 | - | - | - | - | - | - | 5 |
| Consumer participation in using online recommendation agents . . . | 2012 | - | - | - | 1 | - | - | - | - | - | - | - | 1 |
| Internet usage for travei and tourism: The case of Spain | 2011 | - | - | - | - | - | - | - | - | - | 1 | - | 1 |
| lnfluence ofsocial norrns, perceived playfulness and online . . . | 2011 | - | - | - | - | - | 1 | - | - | - | - | - | 1 |
| Total | | - | - | 4 | 3 | 2 | 7 | 7 | 4 | 12 | 3 | - | 39 |

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
