# Peer review of "Consumer Marketing Strategy and E-Commerce in the Last Decade: A Literature Review"

_jtaer, doi:10.3390/jtaer16070164_

Round 1
Reviewer 1 Report
Comments and suggestions for Authors
- Materials and Methods: the presentation of the article in Page 2-3, Please make clear the method and the result of the study that can make reader easy to understand.
- Materials and Methods: page 2, RL 35-36 I don’t see formulating the questions, please make sure! And that related to page 3, RL=82-83, Suggestion that could be possible related to research problem that recover in this study.
- Page 3 RL= 86-88, didn’t see at the figure 1 reaching 245 articles
- Page 4, RL 140, 149 and 151, please make sure the table number.
- Page 4, RL= 145-148 I didn’t see in any table! Could you provide the table on this part? And specially 145 and 148 there are confusion such as 31,25%
- Page 6, RL= 185 and 187 Appendix and In Page 17, RL=743 -746 please check again.
Discussions Page 8. Discussions are fine but in abstract Page 1 RL= 25 wrote Implications for future research but I don’t see the clear of the implication on discussion section.
Please modify clearer on discussion that could provide clear on implications
Please check hyping error such as RL= 316,318,325,338,340,345,353,385,386,391,398,423,429,439,478,480,485,493,496,499,505,516,529,574,576,584, 612,629,678,679,687,695,698, and may in the case with others part please check again.
October 11

Author Response
Reviewer 1:
Dear Reviewer,
After a careful reading of the review and ensuing suggestions, the following changes were performed throughout the article:
Presentation of results were added to the Abstract;
The goal of the research was underscored in the introduction section and the methodology path was emphasized both in the Introduction and Methods sections,
The phrase ‘245 articles’ was erased;
The phrase ‘implications for further research was erased, as well;
Table number was corrected and the confusion of 31.25% clarified;
Formatting was improved
Best regards,
Ricardo Raimundo
Albérico Rosário

Reviewer 2 Report
The article is the result of interesting, current, and useful work. The usefulness of this research can be raised and implicitly the article improved by better clarifying the methodology used and explaining, even briefly, the solution used to analyze the 66 bibliographic resources analyzed (including the 50 articles, 15 contributions to conferences, and a chapter of scientific book).
I think a slight revision of the text could reduce the degree of similarity, which in the case of such research risks being high.
Author Response
Reviewer 2:
Dear Reviewer,
After a careful reading of the review and ensuing suggestions, the following changes were performed throughout the article:
The goal of the research was underscored in the introduction section;
The methodology path was emphasized both in the Introduction and Methods sections, whereas the Methods section was partly rewritten;
Best regards,
Ricardo Raimundo
Albérico Rosário

Reviewer 3 Report
This work contributes to the study of consumer marketing strategy on E-commerce. The work could be of interest to E-commerce researchers, but the theoretical contribution needs to be strengthened and the consumer marketing strategy needs to be better justified.
The following comments are made to guide the authors in strengthening their work:
(1) Please presentation the research results in abstract.
(2) Introduction: The motivation for the study needs to be strengthened. It is not convincing that consumer marketing strategy on E-commerce in need of literature review. Please emphasize in the introduction the importance of the need for a literature review on this topic, such as: Integrated Production and Transportation Scheduling in E-Commerce Supply Chain with Carbon Emission Constraints; The effect of employees' politeness strategy and customer membership on customers' perception of co-recovery and online post-recovery satisfaction. Strengthen statements about the intended theoretical contribution of the work.
(3) The authors do a good job in summarizing the Network of all keywords and Network of Linked Keywords in Fig. 3 and Fig. 4.
(4) Discussion: The title of the article is consumer marketing strategies for e-commerce, but there is no information related to the breakdown of e-commerce marketing strategies in the discussion, which marketing strategies are classified and what are the characteristics of each. The authors need to strengthen the description about the marketing strategy aspects. Rewrite the ideas discussed if necessary.
(5)Discussion: Section 5.1 details the evolution of e-commerce, and it would be clearer if the evolution of e-commerce could be presented in the form of a diagram. In the current exposition, the concept of e-commerce is introduced from its conception to its success, but why the information related to consumer behavior and purchase intention (lines 303-321) is written later, and whether this is related to the evolution of e-commerce, the authors need to clarify which view this section wants to present.
(6) Discussion: Section 5.3 describes the factors influencing consumer behavior and purchase intention, however, there is no summary of existing research on the factors influencing e-business purchase intention, the text just lists different influencing factors in different paragraphs. The authors need to classify the factors influencing e-business purchase intention, and this section needs a summary paragraph.
(7) Discussion: Missing section 5.5 headings.
(8) Please check the formatting issues.
Author Response
Reviewer 3:
Dear Reviewer,
After a careful reading of the review and ensuing suggestions, the following changes were performed throughout the article:
Presentation of results were added to the Abstract;
The goal of the research was underscored in the introduction section;
The way marketing strategies interplay with environmental issues are highlighted in the discussion section, according to the limited information provided by the 66 mentioned articles;
Section 5.1, also includes the evolution of consumer behavior, in parallel with E-commerce;
Section 5.3, also includes a final paragraph;
A heading 5.5. was added;
Formatting was improved
Best regards,
Ricardo Raimundo
Albérico Rosário

Round 2
Reviewer 3 Report
After the first round of revision, this paper have been improved according to the comments. The following issues should be further revised
- some more keywords should be added, such as E-commerce , Systematic Bibliometric Literature Review
- the newest works about Consumer Marketing Strategy and E-commerce should be further reviewed, such as: The effect of employees' politeness strategy and customer membership on customers' perception of co-recovery and online post-recovery satisfaction; An Intelligent Method for Lead User Identification in Customer Collaborative Product Innovation